# High Mechanical Performance Based on Physically Linked Double Network (DN) Hydrogels

**DOI:** 10.3390/ma12203333

**Published:** 2019-10-12

**Authors:** Li Niu, Yutao Zhang, Liyu Shen, Qiuyue Sheng, Shuai Fu, Shiyan Chen, Yun Du, Ying Chen, Yupeng Liu

**Affiliations:** 1Institute of Chemical Industry of Forestry Products, CAF, Nanjing 210042, China; 2Key Laboratory for Organic Electronics & Information Displays and Institute of Advanced Materials, Nanjing University of Post & Telecommunications, Nanjing 210021, China; 3School of Chemistry and Chemical Engineering, Nanjing University, Nanjing 210023, China

**Keywords:** double network hydrogels, hydroxyethyl cellulose, one pot

## Abstract

A new design strategy was proposed to improve the mechanical performance of double network (DN) hydrogels by introducing polyhydroxy compounds into the DN structure and form a physically linked double network through the interaction of hydrogen bonding. Herein, agar/poly(acrylic acid)/hydroxyethyl cellulose composite hydrogels could be prepared by a simple one-pot method. The resulting hydrogels exhibit highly mechanical properties and excellent recoverability, which have potential applications in biomedical fields.

## 1. Introduction 

Unlike most solid materials, hydrogels consist of a hydrophilic polymer network and a large amount of water, the wet soft structure of which allows them to be used as biocompatible materials, such as biosensors [1,2], bio-separators [3,4], drug delivery carriers [5,6,7], tissue engineering scaffolds [8,9,10,11], super capacitor [12,13,14], and intelligent devices [15,16]. Nevertheless, the mechanical properties of conventional chemical cross-linked polymer hydrogels are too weak to be practically applied in some fields [17]. To overcome this shortcoming and expand the application fields of hydrogels, researchers have developed many types of hydrogels with new network structures, such as double-network (DN) hydrogels [18], topological hydrogels [19], nanocomposite hydrogels [20], tetra-arm polymer-based hydrogels [21], magnetic hydrogels [22], double cross-linked hydrogels [23], triblock copolymer hydrogels [24], and hydrogen bonding hydrogels [25]. Among these, the DN hydrogels have been regarded as one of the most excellent hydrogels materials, which can be prepared by combining two interpenetrating polymer networks: a short-chain network as the first network and a long-chain network as the second. The use of this type of interpenetrated structure of the two polymer networks improved and balanced mechanical properties by tuning the intramolecular interactions between two networks [26]. Upon extra-force loading, an amount of the short chains were sacrificed to dissipate energy, and the long chains kept elasticity to retain the whole structure [27], resulting in outstanding fracture stress, critical compression, elastic modulus, and elongation of the DN hydrogels. Recently, reports have focused on various DN hydrogels materials due to their excellent mechanical properties. For instance, the compressive fracture stress of the poly(2-acrylamido-2-methylpropanesulfonate)/polyacrylamide DN hydrogel could achieve 93.5 MPa greater than those of bovine cartilage [28]. The polyacrylamide-chitosan DN hydrogel displayed high tensile strength up to 2 MPa [29]. The poly(vinyl alcohol)-poly(ethylene glycol) DN hydrogel exhibited the shape memory with tensile strength of 0.63 MPa [30]. 

Hydrogen bonding can improve the mechanical properties of hydrogels. In contrast with covalent bonding, the strength of individual hydrogen bonding interaction was relatively weak, but the synergistic interactions among hydrogen bonds could produce strong interactions even over covalent bonding [31]. The composite hydrogels based on hydrogen bonding between polyacrylamide and bacterial cellulose were reported to exhibit good tensile with a breaking elongation of 2200% and a breaking stress of 1.35 MPa [32]. The mechanical properties of polyacrylamide hydrogels were enhanced by hydrogen bonding interaction using graphene oxide [33]. Moreover, hydrogen bonding not only ensures good mechanical properties of the hydrogels but also endow excellent recovery properties of the hydrogel [34]. In our previous work, the lignosulfonate-graft-poly(acrylic acid)/hydroxyethyl cellulose composite hydrogels with semi-interpenetrating networks (semi-IPNs) structure show excellent shape-recovery property owing to the hydrogen bonding interactions between proton-donating Polyacrylic acid (PAA) and proton-accepting hydroxyethyl cellulose (HEC) [35]. 

Normally, the DN hydrogels were prepared in a two-step process: firstly, a single-network hydrogel was prepared. Next, the dried sample was immersed in the second monomer solution until the second monomer completely filled the sample; it was then gelled to form a DN hydrogel. This synthetic method had a long preparation cycle and needed to use a large amount of the second monomer soaking liquid [36]. Chen’s group reported a simple and quick method for the preparation of the Agar/PAM double-network hydrogels in one step and the whole process of preparation only took a few hours [37]. Inspired by this work, we one-pot synthesized agar/poly (acrylic acid)/hydroxyethyl cellulose composite hydrogels (Agar/PAA/HEC DN hydrogels) by introducing polyhydroxy compounds HEC into the DN structure, which physically linked double network by hydrogen bonding. The resulting hydrogels exhibited highly mechanical properties and excellent recoverability. 

## 2. Materials and Methods

### 2.1. Materials

Agar, N,N’-Methylenebisacrylamide (MBAAm) were purchased from Sino pharm Chemical Reagent Co., Ltd. (Shanghai, China); acrylic acid (AA) was purchased from Macklin Chemical Reagent Co., Ltd. (Shanghai, China); hydroxyethyl cellulose (HEC, 5000–6400 mpa.s), 2-hydroxy-4’-(2-hydroxyethoxy)-2-methylpropiophenone were purchased from Aladdin Chemical Reagent Co., Ltd. (Shanghai, China). All other agents were analytical grade and used without further purification.

### 2.2. Synthesis of Agar/PAA/HEC Double Network (DN) Hydrogels 

The agar/PAA/HEC DN hydrogels were synthesized by a one-pot strategy in which 0.1 g of agar was added into 10 mL of H_2_O at 90 °C and stirred until the agar power was dissolved completely and a transparent agar water solution was obtained. Then, 2 g of AA, 5.4 mg of MBAAm, 65.4 mg of 2-hydroxy-4’-(2-hydroxyethoxy)-2-methylpropiophenone, and 0.15 g of HEC were added into the agar solution to dissolve. The obtained solution was cooled to room temperature and subsequently the agar was solidified into gel again. Then, polymerization of AA monomers was carried out under the radiation of UV light (INTELLI-RAY 400, Lamp Intensity 60%, 400 W, Shenzhen, China) for about 50 min. The final hydrogel was immersed into distilled water for 48 h to remove the unreacted monomer and other impurities.

### 2.3. Fourier-Transform Infrared Spectroscopy (FTIR) Test

ATR (attenuated total reflectance)-Fourier-transform infrared spectroscopy (FTIR) of the Agar/PAA/HEC hydrogels was performed on a Nicolet iS10 FT-IR spectrometer (Nicolet iS10, Nicolet, Waltham, MA, USA). The hydrogel was lyophilized to dry gel. The samples were scanned 16 times and the scanning range was 4000–500 cm^−1^.

### 2.4. Morphology Characterization

Morphology characterization was investigated by observing the cross-section of the Agar/PAA/HEC hydrogel using scanning electron microscopy (SEM) (S-3400N Hitachi, Chiyoda, Tokyo, Japan). The hydrogel sample was lyophilized to dry gel. Cryo-fracturing of the frozen specimens was done to obtain a cross-sectional interior of the hydrogels, revealing their interior structure. SEM pictures of the hydrogel were taken after coating with gold nanoparticles under vacuum.

### 2.5. Measurement of Mechanical Properties

The compressive stress–strain measurements were performed using an Instron 5565 electronic universal testing apparatus (Instron Co, Boston, MA, USA) with a 5 kN load cell, a compressive strain rate of 10 mm/min, and no preload at room temperature. The cylindrical hydrogel samples were ~14–16 mm in diameter and 12~15 mm in height. The compressive stress (σ) was estimated as σ = F/πr^2^, where F is the force applied on the hydrogel and r is the initial radius of the sample. The compressive strain (ε), was defined as the change in the height relative to the height of the original sample. 

For tensile test, the measurements also were performed by using an Instron 5565 electronic universal testing apparatus (Instron Co, Boston, MA, USA); a 250 N load cell was used. The hydrogel samples were cut into a dumbbell shape, with a gauge length of 16 mm, a width of 4 mm, and a thickness of ~2–3 mm. Moreover, the stretching rate was 50 mm/min. The cyclic tensile tests were performed immediately following the initial loading. In both loading and unloading, the rate of stretch was kept constant at 100 mm/min. The tensile fracture stress was defined as the stress at the breaking point. Tensile modulus was calculated from the slope of the linear region (ε = ~5–10%) of the stress–strain curves. For the all mechanical testing, a water spray was applied to the sample with a kettle every 2 min to prevent the hydrogel from drying out.

### 2.6. Swelling Properties Test

Swelling studies were carried out by comparing the fully swollen weight and the dry weight of hydrogels. The dried hydrogels (W_1_) were immersed in an excess amount of deionized (DI) water at 25 °C or physiological solution (NaCl 0.9% w/w, NS) at 37 °C until swelling equilibrium was attained. The wet weight of the sample (W_c_) was determined after removing the surface water with filter paper. Equilibrium swelling ratio (Qeq) was determined by the following equation: Qeq = (W_2_ − W_1_)/W_1_ × 100%.

## 3. Results and Discussion

### 3.1. Preparation of the Agar/PAA/HEC DN Hydrogels

The Agar/PAA/HEC hydrogels were prepared through a simple one-pot strategy. The preparation scheme is demonstrated in Figure 1. Firstly, all of the reactants: agar, AA monomer, HEC, UV initiator, and cross linker (MBAAm) were dissolved together in water at 90 °C. Agar is in sol state at 90–95 °C and in gel state at 30–35 °C, resulting from the coil-helix structural transition of the agar molecule between high and low temperatures. Upon cooling, the gelation of agar occurred and built a three-dimensional network, which was the first network of the hydrogel. After photo-initiation, the second network was formed by the free-radical polymerization of the AA monomer under ultraviolet irradiation and the Agar/PAA/HEC hydrogel was successfully synthesized. In this system, in addition to the double network structure, the long-chain polyhydroxy macromolecule HEC acted as a physical cross linker agent to form a special DN network structure based on the hydrogen bonding interaction among the carboxyl of PAA, the hydroxyl of agar, and the hydroxyl of HEC. As shown in Figure 2, in comparison with with pure PAA, the characteristic absorption peaks of C=O of PAA in the Agar/PAA/HEC hydrogels were shifted from 1670 to 1696 cm^−1^, confirming the existing of hydrogen bonding interaction between PAA and HEC. Figure 3 shows the interior morphology of the Agar/PAA/HEC hydrogels measured by SEM. Similar to the Agar/PAA and PAA/HEC hydrogel, the Agar/PAA/HEC hydrogels also have porous structures. However, the pore size of the Agar/PAA/HEC hydrogels was much smaller than that of the other two hydrogels. The hydrogen bonding interaction between the two networks reduced the distance between the polymer chains and made the structure of the hydrogel denser, showing the size of the pore decreasing at the macro. The hydrogen bonding that exists in the structure of the hydrogels gradually dissipates the stress and fracture energy when external forces act on the hydrogels, allowing the hydrogels to withstand high strength deformation. In combination with double network and hydrogen bonding interaction, the Agar/PAA/HEC hydrogels would be expected to have excellent mechanical properties.

### 3.2. The Mechanical Properties of the Agar/PAA/HEC DN Hydrogels

As seen in Figure 4, the Agar/PAA/HEC DN hydrogels could withstand the deformation of bending as seen in Figure 4a, knotting as seen in Figure 4b, compression as seen in Figure 4c, and elongation as seen in Figure 4d, without obvious damage, revealing that they possessed excellent compression, bending, and tensile properties. Firstly, the compression property was evaluated. Figure 5 was a typical compressive stress–strain (σ-ε) curve for the Agar/PAA/HEC DN hydrogels. When the strain reached approximately 80%, the stress increased sharply. One explanation for this occurrence is that, with the increase of strain, some chains were fully compressed, resulting in the rapid increase of stress. When the stress intensity was 15.7 MPa, the hydrogels still did not break and the strain achieved 96%. In contrast, the fracture compressive stress of the ordinary PAA/HEC hydrogels and PAA/Agar DN hydrogels were 0.024 Mpa and 7.1 Mpa, respectively, suggesting that HEC significantly improved the compressive property of the Agar/PAA/HEC DN hydrogels. The recovery performance of the Agar/PAA/HEC DN hydrogels was also studied through an loading-unloading compression experiment. As shown in Figure 6, the cylindrical Agar/PAA/HEC DN hydrogel sample was compressed to 90% of original height and remained intact; after the pressure was released, the original shape of the sample was almost restored. The compressive stress–strain curves of the process were further investigated, as seen in Figure 7. There was a hysteresis in the first loading-unloading cycle, implying that an irreversible fracture event may have occurred at strain. Next, the recovered sample was placed in a moist environment for 2 h and the load-unloading compression curve was measured again. No serious deformation and strength degradation occurred in the hydrogel at a highly set critical strain of 80%, revealing the robust DN structure of the agar and PAA network. This result showed the excellent anti-deformation abilities of the Agar /PAA/HEC DN hydrogels.

The Agar/PAA/HEC DN hydrogels exhibited much higher ductility and positive effect of HEC on the tensile properties of the Agar/PAA/HEC DN hydrogels was well reflected in Table 1 and Table 2. We systematically studied the effects of the contents of HEC on the tensile properties of the Agar/PAA/HEC DN hydrogels. In Table 1, we observed that the tensile strain of the hydrogels (Gel-1–Gel-3) from 15.1 to 19.9 and the tensile stress from 125.5 kPa to 160.2 kPa with the increasing HEC content. It was reasoned that, with the addition of HEC, hydrogen bonding was formed between the HEC and the PAA polymer chain, improving the interaction between two networks by physical crosslinking, effectively promoting the energy dissipation of the hydrogels. When the ratio of HEC/AA was 0.15/2 (g/g), the tensile properties of the hydrogel were optimal. Once the ratio of HEC/AA exceeded 0.15/2 (g/g), the performance of the hydrogel did not continue to improve, potentially because the hydrogen bonding interaction had reached saturation point. On the other hand, as shown in Table 2, the hydrogel prepared by high-viscosity HEC had better performance.

The loading-unloading tensile cycle experiment was an effective method to analyze the internal fracture process of the hydrogels. Figure 8 shows the successive loading-unloading cycles of the Agar/PAA/ HEC DN hydrogels at different strain values, in which a new stretching cycle was performed immediately after the end of the previous cycle. The large hysteresis loops of each cycle suggested that the hydrogels could dissipate energy effectively. It was observed that any two adjacent stress–strain curves shared an overlap region to some extent, which pointed to the self-recovery ability of the hydrogels. The recover ability of the hydrogels may be attributable to the hydrogen bonding between HEC and PAA polymer chains. The hydrogen bonding would rupture upon an action of external force, and reformed after the external force was removed.

### 3.3. Swelling Properties of the Agar/PAA/HEC DN Hydrogels

The swelling property of the hydrogels has many applications, which are mainly determined by various functional groups inside the hydrogels. The content of HEC in the Agar/PAA/HEC DN hydrogels plays an important role in the swelling behaviors of the hydrogels. Figure 9 shows that the swelling rate of the hydrogels reduced as the HEC content increased. In physiological saline, the sample was slightly more swollen due to the presence of salt. The carboxyl groups in the PAA determined the swelling rate of the hydrogels; however, with the addition of HEC, the formed hydrogen bonding inhibited the swelling of the hydrogels.

## 4. Conclusions

We have successfully fabricated a new physically linked double network Agar/PAA/HEC hydrogel using a one-pot method by the hydrogen bonding interaction between proton-donating PAA and proton-accepting HEC. The Agar/PAA/HEC hydrogel has very high compression and tensile strength as well as high recoverability, which may be due to the synergy of the network structure and the reversible energy dissipation mechanism of the hydrogen bonding. The optical mass ratio of HEC/AA for the hydrogel is 0.15/2 (g/g). In consideration of the advantages of good mechanical properties as well as one-pot synthetic methods, the agar/PAA/HEC hydrogel has many potentials in biomedical applications, such as tissue engineering scaffolds and artificial muscles.

## Figures and Tables

**Figure 1 materials-12-03333-f001:**
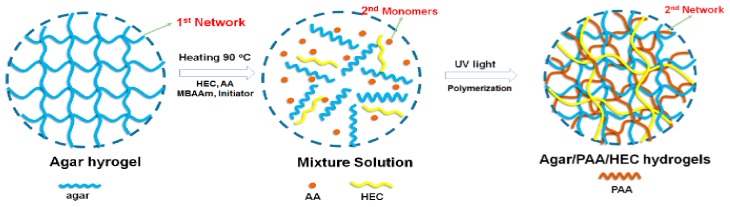
Scheme of the synthesis of the Agar/PAA/hydroxyethyl cellulose (HEC) hydrogels.

**Figure 2 materials-12-03333-f002:**
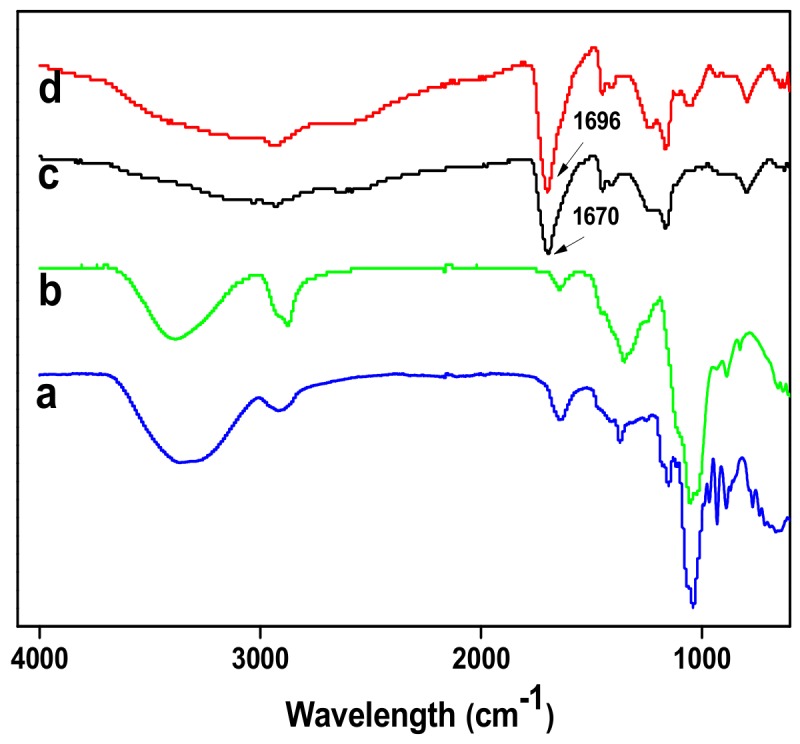
Attenuated total reflectance (ATR)-Fourier-transform infrared spectroscopy (FTIR) spectra of (**a**) Agar; (**b**) HEC; (**c**) PAA and (**d**) Agar/PAA/HEC hydrogel.

**Figure 3 materials-12-03333-f003:**
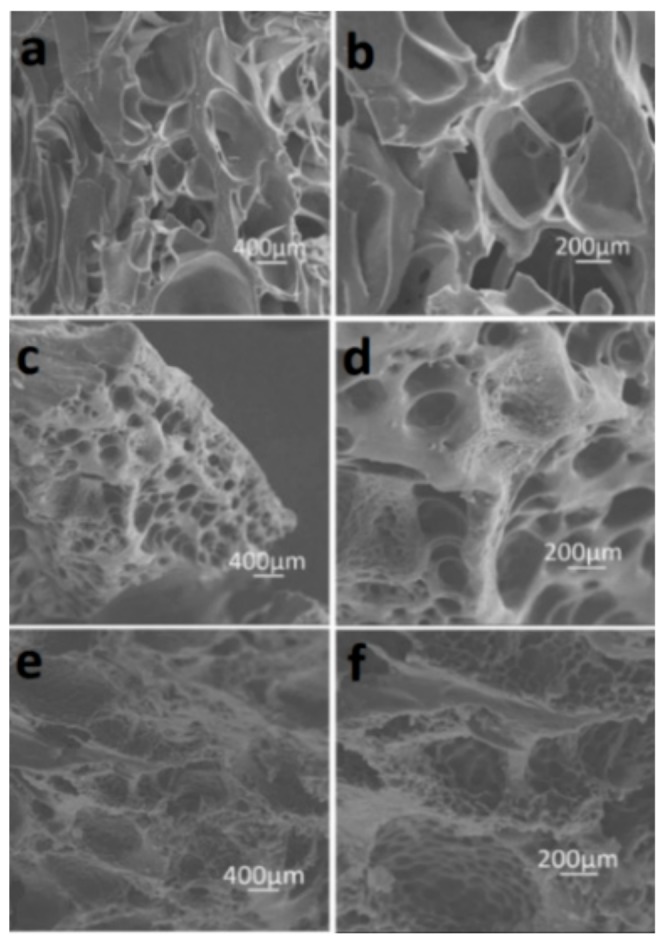
Scanning electron microscopy (SEM) images of hydrogels: (**a**,**b**) Agar/PAA hydrogel; (**c**,**d**) PAA/HEC hydrogel; (**e**,**f**) Agar/PAA/HEC hydrogel.

**Figure 4 materials-12-03333-f004:**
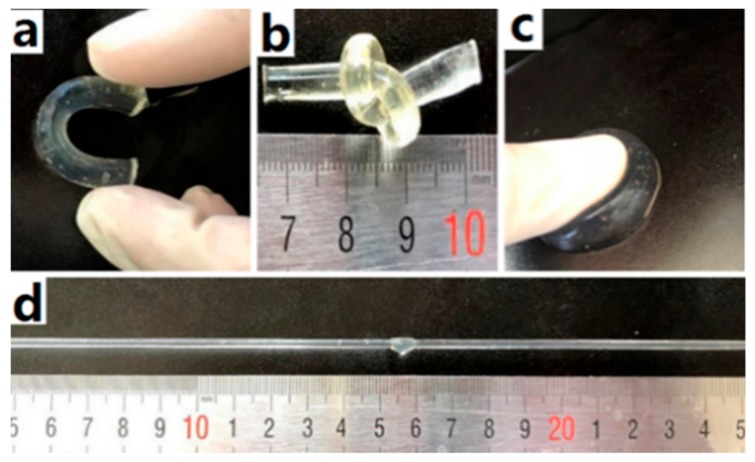
Agar/PAA/HEC hydrogels show extraordinary mechanical: (**a**) Bending; (**b**) Knotting; (**c**) Compression; (**d**) Elongation.

**Figure 5 materials-12-03333-f005:**
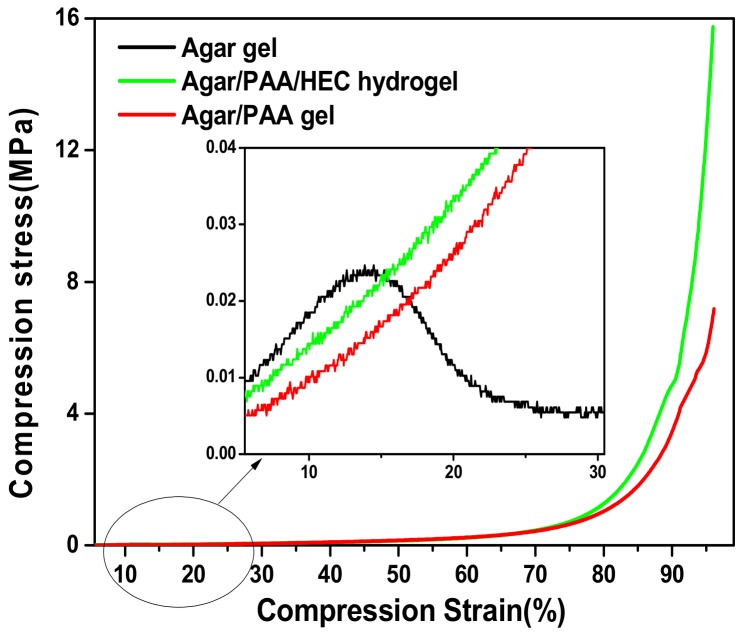
Compression stress–strain curves of different hydrogels.

**Figure 6 materials-12-03333-f006:**
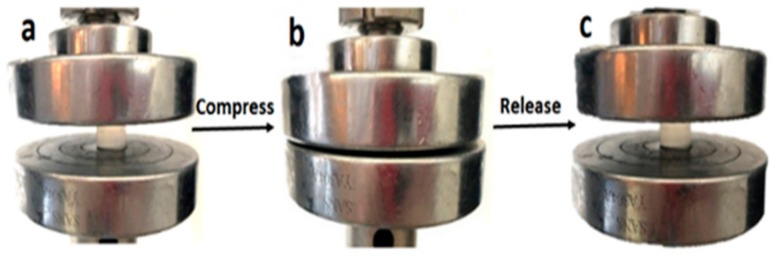
Photographs of the Agar/PAA/HEC DN hydrogel under compression.

**Figure 7 materials-12-03333-f007:**
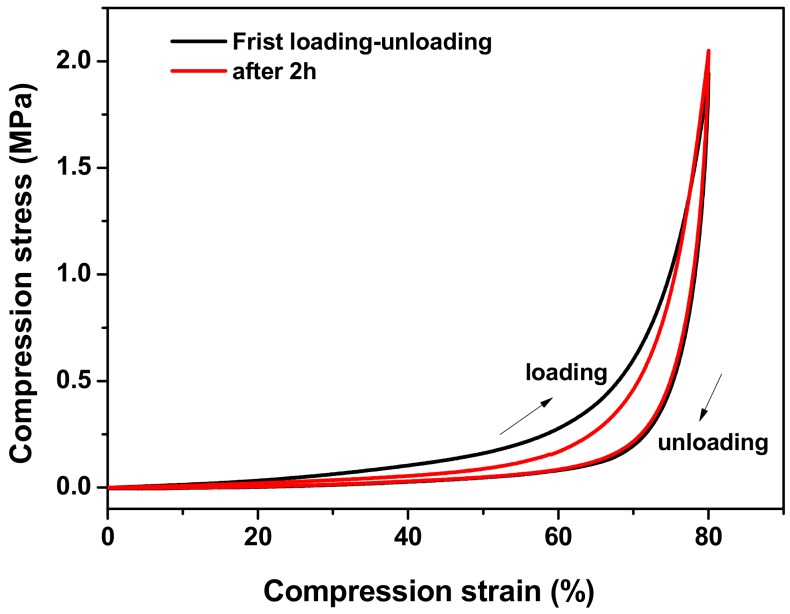
Stress–strain curves of loading-unloading cycles.

**Figure 8 materials-12-03333-f008:**
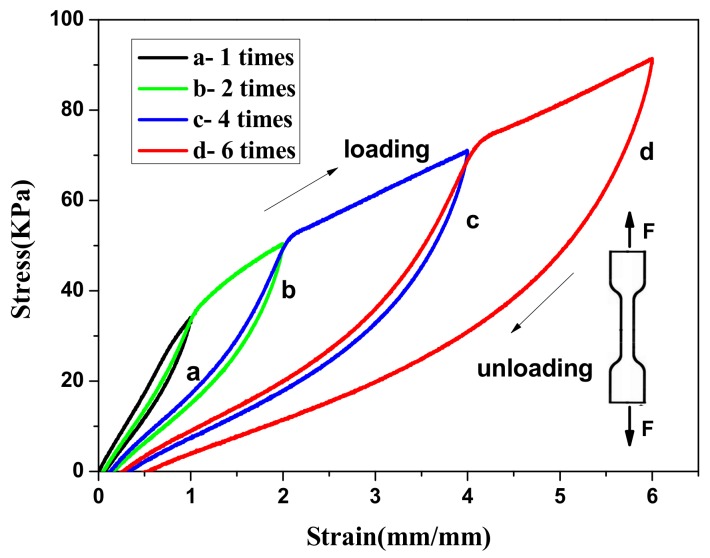
Stress–strain curves during loading-unloading cycles at different critical compression strains of the Agar/PAA/HEC hydrogel.

**Figure 9 materials-12-03333-f009:**
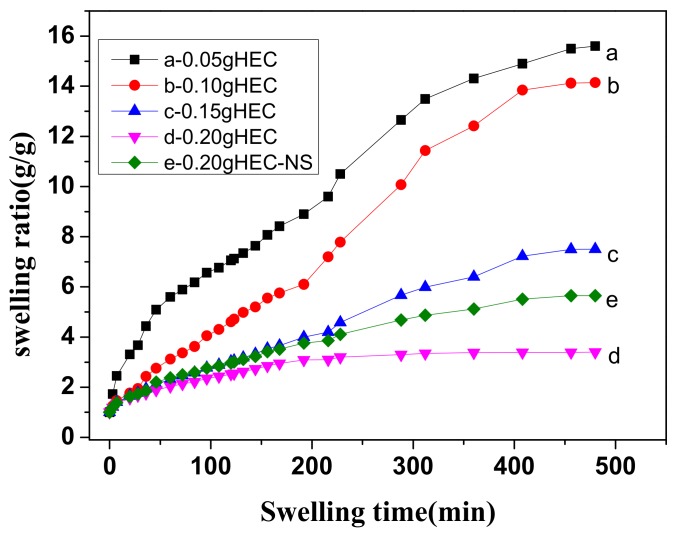
Swelling rate of the different hydrogels (**a**) Gel-1 in DI water; (**b**) Gel-2 in DI water; (**c**) Gel-3 in DI water; (**d**) Gel-4 in DI water and (**e**) Gel-4 in salt solution.

**Table 1 materials-12-03333-t001:** Tensile properties of the Agar/PAA/HEC DN hydrogels with different HEC contents.

Sample	Synthesis		Tensile Test		
	HEC/AA/H_2_O (g/g/g)	HEC viscosity (mpa.s, 25 °C)	Strain at fracture (mm/mm)	Stress at fracture (kPa)	Tensile modulus (kPa)
Gel-1	0.05/2/10	5000–6400	15.1	125.5	4.5
Gel-2	0.10/2/10	5000–6400	16.7	147.6	18.5
Gel-3	0.15/2/10	5000–6400	19.9	160.2	21.9
Gel-4	0.2/2/10	5000–6400	14.6	99.3	9.3

**Table 2 materials-12-03333-t002:** The tensile properties of the Agar/PAA/HEC DN hydrogels with different HEC viscosity.

Sample	Synthesis		Tensile Test		
	HEC/AA/H_2_O (g/g/g)	HEC viscosity (mpa.s, 25 °C)	Strain at fracture (mm/mm)	Stress at fracture (kPa)	Tensile modulus (kPa)
Gel-5	0.15/2/10	80–125	8.6	138.0	27.4
Gel-6	0.15/2/10	1000–1500	12.0	131.4	22.1
Gel-7	0.15/2/10	5000–6400	19.9	160.2	21.9

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
