# Peer review of "High Mechanical Performance Based on Physically Linked Double Network (DN) Hydrogels"

_materials, 2019, doi:10.3390/ma12203333_

Round 1
Reviewer 1 Report
The authors show the preparation lf agar/polyacrylic acid/hydroxyethyl cellulose hydrogels. The work is consistent and the results are clearly shown. However some minor revisions should be made:
in the materials and methods section please include more details about the polymers use, which is the molecular weigth of HEC?
it seems that during mechanical testing the hydrogels were not immersed in water. How did the authors prevent drying of the hydrogels? Were the dimensions of the samples measured before and after the tests? Was there any weigth loss of the samples?
line 188 page 6 please explain why the performance of the hydrogel was not improved further when increasing the ratio of HEC?
in the discussion of the mechanical tests results please compare to the properties of similar hydrogels. Please provide the results for agar/paa gel for comparison
the hysteresis loop in stress-strain curves could be due to hydrogel drying, please discuss this point
in figure 8 what are a,b,c and d reffering to? Include it in the figure legend
figure 9 is missing
page 7 line 210 please rephrase the sentence, “as the Hec contrnt” a verb is missing
Author Response
Comments from the Reviewer#1:
1.in the materials and methods section please include more details about the polymers use, which is the molecular weight of HEC?
Response: Hydroxyethyl-cellulose is a network-forming polymer, the viscosity of which is 5000-6400 mpa.s in 25℃。(Please see the attachment, red highlighting, line 70 page 2 )
2.it seems that during mechanical testing the hydrogels were not immersed in water. How did the authors prevent drying of the hydrogels? Were the dimensions of the samples measured before and after the tests? Was there any weight loss of the samples?
Response: The mechanical testing was carried out at 25 ° C and the test time of the sample was very short, about 1 to 2 minutes. During this test time, the hydrogel stayed dry and the moisture evaporation of the hydrogel could be negligible. For longer mechanical testing, a water spray was applied to the sample with a kettle every 2 minutes to prevent the hydrogel from drying out. Before and after the hydrogel mechanical tests, the samples would be deformed or even broken, but the mass before and after was not lost. (See red highlighting, line 108 page 3)
3.line 188 page 6 please explain why the performance of the hydrogel was not improved further when increasing the ratio of HEC?
Response: It was reasoned that with the addition of HEC, hydrogen bonding was formed between the HEC and the PAA polymer chain, improving the interaction between two networks by physical crosslinking, effectively promoting the energy dissipation of the hydrogels. When the ratio of HEC/AA was 0.15/2 (g/g), the tensile properties of the hydrogel were optimal. Once the ratio of HEC/AA exceeded 0.15/2 (g/g), the performance of the hydrogel was not improved further. This may be due to hydrogen bonding interaction had reached saturation point. (See red highlighting, line 192 page 6)
4.in the discussion of the mechanical tests results please compare to the properties of similar hydrogels. Please provide the results for agar/paa gel for comparison?
Response: Figure 5 was a typical compressive stress–strain (σ-ε) curve for the Agar/PAA/HEC DN hydrogels. When the strain reached approximately 80%, the stress increased sharply, which could be explained that with the increase of strain, some chains were fully compressed, resulting in the rapid increase of stress. When the stress intensity was 15.7 MPa, the hydrogels still did not break and the strain achieved 96%, while the fracture compressive stress of the ordinary PAA/HEC hydrogels and PAA/Agar DN hydrogels were 0.024 Mpa and 7.1Mpa. (See red highlighting, line 154 page 5)
5. the hysteresis loop in stress-strain curves could be due to hydrogel drying, please discuss this point?
Response: We have done some measures in the experiment to prevent the hydrogel from drying, and the quality of the hydrogel did not change before and after the hydrogel test, indicating that the hydrogel did not dry, so this curve does not have the effect of hydrogel drying.
6. in figure 8 what are a,b,c and d reffering to? Include it in the figure legend?
Response: A legend has been added in Figure 8. and a,b,c,d are explained. (See figure 8, line 206 page 8)
7. Figure 9 is missing
Response: Figure 9 has been added. (See figure 9, line 218 page 9)
8. Page 7 line 210 please rephrase the sentence, “as the Hec contrnt” a verb is missing?
Response: Page 7 line 210 please rephrase the sentence as ‘Figure 9 showed that the swelling rate of the hydrogels reduced as the HEC content increases.’. (See red highlighting, line 214 page 8)

Reviewer 2 Report
The manuscript entitled “Highly Mechanical Performance based on Physically Linked Double Network (DN) Hydrogel” results well written, structured and discussed. Overall, the manuscript is satisfactory. However, few major revisions must be made from the authors in order to make the manuscript suitable for publication in Materials.
1) Line 85: Please, specify the acronym ATR.
2) Line 112: swelling tests have been performed in deionized water. However, since it is well known that the saline concentration hugely influences the swelling behaviour of hydrogels and, especially if hydrogels are intended to be used in biomedical field, I think it is necessary to perform the swelling tests in physiological solution (NaCl 0.9% w/w) at 37°C, rather than in deionized water at 25°C.
3) Lines 206-212: the manuscript is missing in Figure 9. Please, add this figure and comment the results of swelling in more detail, also considering my advice to carry out such measurements in a more appropriate medium, such as physiological solution.
Author Response
Comments from the Reviewer#2:
1) Line 85: Please, specify the acronym ATR.
Response: ATR means attenuated total reflectance, which was added to line 85. (Please see the attachment, red highlighting, line 84 page 2)
2) Line 112: swelling tests have been performed in deionized water. However, since it is well known that the saline concentration hugely influences the swelling behaviour of hydrogels and, especially if hydrogels are intended to be used in biomedical field, I think it is necessary to perform the swelling tests in physiological solution (NaCl 0.9% w/w) at 37°C, rather than in deionized water at 25°C. (See red highlighting, line 114 page 3)
Response: The opinions of the experts are very reasonable. We also conducted a swelling tests on the samples in physiological solution (NaCl 0.9% w/w, NS) at 37°C
3) Lines 206-212: the manuscript is missing in Figure 9. Please, add this figure and comment the results of swelling in more detail, also considering my advice to carry out such measurements in a more appropriate medium, such as physiological solution.
Response: Figure 9 has been added and the swelling tests in physiological solution (NaCl 0.9% w/w, NS) at 37°C are added. (See figure 9, line 218 page 9)

Round 2
Reviewer 2 Report
The manuscript has been adequately revised and now it can be accepted for publication in Materials journal.